# Precise Molecular Engineering of Type I Photosensitizer with Aggregation-Induced Emission for Image-Guided Photodynamic Eradication of Biofilm

**DOI:** 10.3390/molecules28145368

**Published:** 2023-07-12

**Authors:** Jinghong Shi, Yucheng Wang, Wei He, Ziyue Ye, Mengli Liu, Zheng Zhao, Jacky Wing Yip Lam, Pengfei Zhang, Ryan Tsz Kin Kwok, Ben Zhong Tang

**Affiliations:** 1Department of Chemistry, Hong Kong Branch of Chinese National Engineering Research Center for Tissue Restoration and Reconstruction, The Hong Kong University of Science and Technology, Clear Water Bay, Kowloon, Hong Kong, China; 2School of Science and Engineering, Shenzhen Key Laboratory of Functional Aggregate Materials, The Chinese University of Hong Kong, Shenzhen 518172, China; 3HKUST-Shenzhen Research Institute, South Area Hi-Tech Park, Nanshan, Shenzhen 518057, China; 4Shenzhen Key Laboratory for Molecular Imaging, CAS Key Lab for Health Informatics, Shenzhen Institutes of Advanced Technology, Chinese Academy of Sciences, Shenzhen 518055, China

**Keywords:** aggregation-induced emission, type I photosensitizer, biofilm, bacterial imaging, anti-biofilm materials

## Abstract

Biofilm-associated infections exert more severe and harmful attacks on human health since they can accelerate the generation and development of the antibiotic resistance of the embedded bacteria. Anti-biofilm materials and techniques that can eliminate biofilms effectively are in urgent demand. Therefore, we designed a type I photosensitizer (TTTDM) with an aggregation-induced emission (AIE) property and used F-127 to encapsulate the TTTDM into nanoparticles (F-127 AIE NPs). The NPs exhibit highly efficient ROS generation by enhancing intramolecular D–A interaction and confining molecular non-radiative transitions. Furthermore, the NPs can sufficiently penetrate the biofilm matrix and then detect and eliminate mature bacterial biofilms upon white light irradiation. This strategy holds great promise for the rapid detection and eradication of bacterial biofilms.

## 1. Introduction

Bacteria are ubiquitous and they hold strong connections with our health. Some bacteria can cause infections [1,2]. Pathogenic bacterial infections are seriously threatening to public health. Moreover, many infections can also indirectly promote the occurrence and progression of other diseases, such as cancer and tuberculosis [3,4,5]. Notably, bacteria can self-aggregate and form an interactable community called a biofilm [6,7,8,9]. Biofilms refer to the intricate three-dimensional aggregation of bacteria attached to a surface and that are buried inflexibly in an extracellular polymeric substance (EPS) matrix [10]. Generally, the EPS matrix is composed of secreted polymers such as polysaccharides, extracellular DNA (e-DNA), proteins, and amyloidogenic proteins [11,12,13]. Bacterial biofilms can be formed in diverse environments, and biofilm-associated infections exert more severe and harmful attacks on human health since they can accelerate the generation and development of the antibiotic resistance of the embedded bacteria [14,15]. In addition, the biofilm matrix offers a protective layer for the bacteria from pH fluctuations, nutrient deficiency, and mechanical forces, leading to superbug concerns [16,17,18]. Therefore, anti-biofilm materials and techniques are in urgent demand.

During the past decades, abundant anti-biofilm strategies have been proposed, investigated, and developed. Antibiotics are the most frequently and widely adopted strategy for the treatment of biofilm-related infections. However, compared with planktonic bacteria, an up to 1000-fold antibiotic dosage is required to eradicate the bacterial cells swathed in the biofilm [19]. And the extensive use of antibiotics has led to more severe multi-drug resistance generation, leading to a shortage of applicable antibiotic pipelines [20,21]. Consequently, other non-antibiotic anti-biofilm strategies have been developed, such as cationic polymer brushes [22], herbal active compounds [23], proteases and DNase [24], antimicrobial peptide (AMP) [25], and photoactive materials [26]. Unlike antibiotics, instead of targeting a specific metabolic pathway, those antimicrobial agents can attack multiple sites on bacterial structures [27,28]. They can kill bacteria through blocking DNA replication, altering bacterial gene expression, denaturing enzymes, or damaging bacterial membranes. Recently, extracts from plants were reported to regulate biofilm formation and break down existing biofilms by inhibiting the quorum-sensing pathway [29]. However, the molecular structures of plant-derived extracts with anti-biofilm activity are hard to identify and hinder further applications. Some enzymes can be utilized as a certain biofilm-dispersing agent such as polysacchradelyases and DNases that are able to disrupt extracellular polysaccharide substances and digest DNA in the matrix [30,31]. However, it is still challenging for enzymes to differentiate bacterial infection sites from normal living tissues. Antimicrobial peptide and polymer brushes can be utilized for fighting against bacteria through the membrane disruption mechanism of their cationic portions [32]. Although these novel materials played a significant role in defeating biofilms, they suffered from tedious synthesis, complicated protocols, and limited antifouling abilities. Therefore, anti-biofilm materials with simpler preparation procedures, higher stabilities in physiological environments, and universal anti-biofilm abilities are required.

Photodynamic therapy (PDT) is an emerging clinical treatment method owing to its superior merits such as its spatial-temporal controllability, high efficiency, and noninvasiveness [33]. The PDT technique also shows promising treatment abilities towards drug-resistant bacteria and biofilms. Basically, reactive oxygen species (ROS) can be generated by photosensitizers upon light irradiation and can then kill bacteria or destroy biofilms by interacting with the bacteria or key components and causing oxidative damage to the vital cellular components of the bacteria [22]. Thus, the efficiency of photosensitizers plays an important role in PDT. However, conventional photosensitizers such as porphyrins and rose bengal suffer from the aggregation-caused quenching (ACQ) effect due to their strong π–π interaction in aggregates [34]. Because strong π–π stacking can greatly accelerate the non-radiative decay of the exciton, it thereby reduces the efficiency of intersystem crossing (ISC) and ROS generation. The ACQ phenomenon can not only lead to reduced fluorescence signals but also cause compromised photosensitization efficiencies at high concentrations or aggregate states. Fortunately, photosensitizers with aggregation-induced emission (AIE) characteristics can address these issues [35]. Under light irradiation, the AIE PSs, in their triplet excited state, transfer energy to the surrounding O_2_ to generate singlet oxygen [36]. The ^1^O_2_ production efficiency is highly dependent on the amount of O_2_ in the surrounding area, thus the PDT outcome of the type II PSs is extremely hindered by tissue oxygenation. On the contrary, the type I PSs produce free radicals in a different mechanism. A hydrogen or electron transfer process occurs between the triplet excited states of PSs and the substrate, which can interact instantaneously with O_2_, or other molecules to produce superoxide radicals, hydroxyl radicals, and other oxygen-related radicals [37,38,39]. In other words, type I PSs show that less oxygen-dependent properties are better alternatives for anti-biofilm treatment.

Furthermore, an efficient ISC process is widely recognized as one of the key factors for effective ROS generation. The construction of molecules with high electron donor (D)–acceptor (A) strength usually represents an effective strategy to accelerate the ISC process owing to the decreased singlet–triplet energy gap caused by separating the highest occupied molecular orbital (HOMO) and the lowest unoccupied molecular orbital (LUMO) distribution [40]. In this contribution, we designed and synthesized a type I AIE PS by introducing strong electron donor and electron acceptor pairs with a π-bridge. The strong donor and acceptor system can greatly separate the HOMO–LUMO distribution to decrease Δ_Est_. Meanwhile, the strong electron withdrawing unit can also significantly shift the molecule emission to the near-infrared region. To enhance the penetration ability of the hydrophobic molecule into the biofilm, we encapsulated the AIE PSs using a biocompatible polymer F-127. Moreover, the restriction of intramolecular motion in the aggregate state can significantly enhance ROS generation efficiency by prohibiting energy dissipation. F-127 AIE NPs showed high ROS generation ability compared with the commercial photosensitizer rose bengal. In our work, F-127-encapsulated AIE NPs can also act as a fluorescent biomarker for biofilm. Due to their electrically neutral properties, F-127 AIE NPs can interact better with the biofilm matrix compared with individual bacteria. With the help of F-127 AIE NPs’ initial photosensitizing ability, the labeled biofilm can be ablated upon light irradiation through PDT (Figure 1).

## 2. Result and Discussion

### 2.1. Molecular Design and Synthesis

In general, the integration of strong electron donating–accepting groups into the conjugated fluorophores could greatly facilitate intramolecular charge transfer (ICT) and enhance D–A strength, thus resulting in high ROS generation efficiency as well as longer absorption and emission wavelengths [41]. As depicted in Figure 1, the target compound possessed a conjugated D–A structure. The synthetic route of the designed compound was presented in Figure 1. The tetraphenylethene derivative 1-(4-bromophenyl)-2,2-bis(4-methoxyphenyl)-1-phenylethene was utilized as the starting material. Compound **1** was synthesized using Buchwald–Hartwig coupling with aniline using Pd_2_(dba)_3_ and tri-*tert*-butylphosphine as a catalyst/ligand combination in toluene with an 80% yield. Then, the bis-arylamine **1** reacted with 1-bromo-4-iodobenzene, copper(I) iodide, and potassium hydroxide in C-N coupling, delivering compound **2** in a 79% yield. Next, compound **2** underwent the Suzuki reaction with 5-formyl-2-thiopheneboronic acid to give compound **3** in a yield of 90%. By condensing compound **3** with 2-(3,5,5-trimethyl-2-cyclohexen-1-ylidene) in ethanol, the target molecule **4** (TTTDM) was obtained in a good yield of 87%. The structure of the TTTDM and intermediates were fully characterized using ^1^H NMR, high-resolution mass spectra (HRMS) in the Appendix A.

### 2.2. Photophysical Properties of TTTDM and F-127 AIE Nanoparticles

Firstly, the photophysical properties of TTTDM were investigated. Benefiting from the enhancement of D–A strength, emission wavelengths of TTTDM reached the near-infrared region (Figure 2A,B). The maximum emission peaked at 689 nm, implying the possibility of near infrared (NIR) imaging. In addition, HOMO electron densities of TTTDM were primarily distributed in methoxy-TPE and benzenamine moieties, while the LUMO was localized over the other thiophene and cyano fragments (Appendix A), indicating the separated HOMO–LUMO and typical D–A structural features.

The AIE characteristics of the compound were further studied by monitoring their emission behaviors in THF/H_2_O mixtures with different water fractions. As shown in Figure 2D–F, TTTDM only gave a weak fluorescence in a pure THF solution. When the water fraction (*f*_w_) was increased from 0 to 60%, emission wavelengths were slightly red-shifted. This could be attributed to an increase in the solvent polarity and TICT mechanism. When *f*_w_ reached over 60%, the fluorescence emission intensity enhanced dramatically, indicating the typical AIE feature of this compound.

To improve its water solubility and biocompatibility, the relatively hydrophobic TTTDM was encapsulated into nanoparticles via the nano-precipitation method using an amphiphilic Pluronic F-127. Using dynamic light scattering (DLS), the average hydrodynamic diameter of F-127 AIE nanoparticles was determined to be ~108 nm (Figure 2C), which is beneficial to efficient accumulation at the biofilm matrix for fluorescence imaging and anti-biofilm PDT. Likewise, the F-127-encapsulated nanoparticles exhibited uniform morphology as demonstrated in TEM images shown in Figure 2C. What is more, the zeta potential was 0.3 mV, which is close to electrical neutrality.

### 2.3. ROS Generation Ability of F-127 AIE Nanoparticles

To verify our design strategy, the type I ROS generation capacities of TTTDM were assessed. First of all, we evaluated the total ROS generation abilities of the commercial photosensitizer rose bengal (RB) and F-127 AIE NPs. Then, a classic ROS indicator dichlorodihydrofluorescein (DCFH), which can easily be oxidized using any type of ROS to fluorescent dichlorofluorescein (DCF), was applied. In order to make the results more reliable, we used the commercially available RB as a reference and also set a blank control group with DCFH (5 μM) alone. As shown in Figure 3A, under the irradiation of white light, the fluorescence intensity (525 nm) of DCF showed rapid growth in the presence of F-127 AIE NPs (1 μM), which indicates the fast generation of ROS. After 14 min, the emission intensity of DCF reached a plateau which was about 420-fold higher than the initial intensity. In contrast, the emission intensity of DCF in the presence of RB (1 μM) also showed a rapid enhancement in the first 2 min of white light eradiation, but after that, the upward trend slowed down significantly and eventually stalled at nearly 120-fold higher than the initial fluorescence intensity. Meanwhile, the irradiated solution with DCFH alone showed no apparent enhancement in the emission intensity. The above results indicate that F-127 AIE NPs showed superior total ROS production efficiency when compared with RB. In our theoretical calculation result, the calculated singlet–triplet energy gap of TTTDM is relatively small. It is believed that the donor–acceptor strength contributed to the significant generation of ROS (Appendix A). Furthermore, the ROS type that was generated by F-127 AIE NPs was determined by other ROS indicators, including hydroxyphenyl fluorescein (HPF) as an •OH indicator, dihydrorhodamine 123 (DHR123) as an O_2_^•−^ indicator, and 9,10-anthracenediyl bis(methylene)dimalonic acid (ABDA) as an ^1^O_2_ indicator. The HPF can be served as an •OH indicator as its reaction product emits green fluorescence centered at 515 nm. As shown in Figure 3B, HPF alone has almost no fluorescence intensity enhancement at 515 nm after white light irradiation. However, the emission intensity of HPF solution shows obvious enhancement in the presence of AIE NPs, which certifies that F-127 AIE NPs have the capability to produce •OH via a type I process. Meanwhile, Figure 3C indicates that F-127 AIE NPs are also capable of generating O_2_•^−^ through a type I process with a superior production rate and total quantity when compared with RB. Furthermore, to exclude the contribution of ^1^O_2_ to ROS production, ABDA was used to selectively detect ^1^O_2_. As shown in Figure 3D, under white light irradiation with different times, the absorption peak at 378 nm of ABDA slightly decreased with F-127 AIE NPs or nothing added, revealing the poor ^1^O_2_ generation ability. Taken together, F-127 AIE NPs have superior total ROS generation efficiency than RB through the type I pathway, which makes them a more competitive photosensitizer when utilized in hypoxia environments.

### 2.4. Bacteria and Biofilm Imaging

Aiming to investigate the imaging ability of F-127 AIE NPs on planktonic bacteria using confocal laser scanning microscopy (CLSM), *P. aeruginosa* was employed as the preliminary representative bacteria. Bacteria-infected Luria–Bertani (LB) broth medium was first added to a confocal dish with a glass coverslip at the bottom, providing a platform for successive bacterial attachment, bacterial auto-aggregation, and biofilm formation. After a certain incubation period, bacteria were stained with 80 μM F-127 AIE NPs for 60 min and imaged under a confocal microscope. As shown in Figure 4, red emission was observed over the dense *P. aeruginosa* plaque, while very weak red emission was found for bacterial individuals. What is more, the bacteria plaque morphology can be observed in 3D confocal images with the help of F-127 AIE NPs staining. The appearance of bacterial plaque clumps is indicative of biofilm formation, which is regarded as a dynamic matrix layer consisting of EPS; mainly, extracellular polysaccharides. Once F-127 AIE NPs were attached to the biofilm surface, they then gradually diffused into the matrix environment, as reflected by the fluorescent signal in Figure 4. In this case, F-127 AIE NPs were believed to be trapped in the sticky dynamic bacterial-synthesized polymeric film, thus labeling the biofilm area with red emission.

### 2.5. Anti-Biofilm Treatment

Encouraged by excellent ROS generation and selective targeting ability, the killing capability of F-127 AIE NPs to *P. aeruginosa* biofilm was further examined. As presented in Figure 5, after the treatment of *P. aeruginosa* with F-127 AIE NPs for 6 h, bacterial aggregation with bright red emission was observed, while some individual bacteria were not labeled. Meanwhile, it was found that the size of bacterial clumps enlarged with an increasing F-127 AIE NP dose from 10 to 80 μM. Although bacterial auto-aggregation processes were still poorly understood, it was generally regarded as a phenomenon that occurred under external conditional changes, such as temperature fluctuation, nutrient starvation, and other external stress [42]. The results suggested F-127 AIE NPs could act as an external substance that drives the bacteria to initiate a series of protective mechanisms through the auto-aggregation process. Inspired by the excellent performance of F-127 AIE NPs in detecting and grouping bacteria, the anti-biofilm performance was further explored. An amount of 80 μM of F-127 AIE NPs were incubated with *P. aeruginosa* mature biofilms and then treated with and without white light (40 mW cm^−2^) for 10 min. Crystal violet (CV) staining was employed to quantify the biomass. CV is a basic dye, and it binds to negatively charged surface molecules and polysaccharides in the extracellular matrix. As the results show in Figure 6, F-127 AIE NPs together with white light irradiation could significantly reduce the biomass of a biofilm when compared with NPs alone.

Following the excellent performance of F-127 AIE NPs in biofilm imaging, CLSM was also applied to confirm the biofilm inhibition ability with F-127 AIE NPs. As illustrated in Figure 7, a layer of biofilm with red fluorescence from F-127 AIE NPs is observed for the dark group. However, merely a few cluster molecules are observed in the CLSM image in the light group. Such obvious comparison demonstrated F-127 AIE NPs can efficiently inhibit biofilm formation under phototoxicity. In addition, as shown in Appendix A, dispersing the biofilms of the light group and dark group in PBS can also clearly highlight the biofilm of the light group breaking up, while the biofilm of the dark group is still in a turbid state. As shown in Appendix A, scanning electron microscopy (SEM) images revealed biofilm morphology changes before and after photodynamic treatment (Appendix A). The SEM results reflect that the treated bacteria exhibited rupture and uneven morphology in comparison with the bacterial control group with a well-defined bacterial shape and borders.

## 3. Conclusions

In this work, we utilized a facile design strategy of intramolecular D–A interaction enhancement designed TTTDM, which resulted in desirable efficient type I ROS generation and fluorescence in the NIR region. The strategy solves the problem of type II PSs for hypoxia-overcoming PDT, which is beneficial for biofilm eradiation. More importantly, the F-127 AIE NPs can successfully stain biofilms and boost their auto-aggregation process. With the help of good ROS generation ability, the labeled biofilm was ablated upon light irradiation. As biofilm-embedded bacteria have strong drug resistance, this work provides a sufficient method that can detect and eliminate biofilm-associated infection.

## 4. Experimental Sessions

### 4.1. Materials

All the solvents and reagents used in this work were of analytical grade. The biological chemical reagents 9, 10-anthracenediyl-bis (methylene)-dimalonic acid (ABDA), 2′,7′-dichlorodihydrofluorescein diacetate (H2DCF-DA), and crystal violet were offered from aladdin Co. 2-(3,5,5-Trimethyl-2-cyclohexen-1-ylidene)propanedinitrile was purchased from Macklin Biochemical Co., Ltd. (Shanghai, China) 1,1′-[2-(4-Bromophenyl)-2-phenylethenylidene]bis[4-methoxybenzene] and Pluronic F-127 were customized from Bidepharm Co., Ltd. (Shanghai, China) Aniline, Pd_2_(dba)_3_, Pd(dppf)Cl_2_, P(t-Bu)_3_, 1-bromo-4-iodobenzene, malononitrile, and the solvents were purchased from J&K Co., Ltd. (Hong Kong, China)

### 4.2. Instrument and Characterization

^1^H and ^13^C NMR spectra were measured on a Bruker AVANCE NEO 500 NMR spectrometer using CDCl_3_ as a solvent. High-resolution mass spectra were obtained using a Xevo G2-XS Q-Tof mass spectrometer. UV–Vis absorption spectra were measured on a PerkinElmer LAMBDA 465 spectrophotometer. Fluorescence emission spectra were recorded using a Techcomp FL970 spectrofluorometer. The hydrodynamic diameter of F-127 AIE nanoparticles were determined through dynamic light scattering on a particle size analyzer Malvern Zetasizer Nano ZSP. Transmission electron microscopy (TEM) investigations were carried out on a JEM-F200 microscope.

### 4.3. Synthesis and Purification

#### 4.3.1. Synthesis of **1**

A solution of 1,1′-[2-(4-bromophenyl)-2-phenylethenylidene]bis[4-methoxybenzene] (2.35 g, 5 mmol), substituted aniline (6.5 mmol), tri-tert-butylphosphine (16.2 mg, 0.08 mmol), Pd_2_(dba)_3_ (64 mg, 0.07 mmol), and sodium tert-butoxide (625 mg, 6.5 mmol) was refluxed under nitrogen in dry toluene (30 mL) at 110 °C for 24 h. After cooling to room temperature, the solvent was removed through evaporation under reduced pressure. Water (30 mL) and chloroform (200 mL) were then added. The organic layer was separated and washed with brine, dried over anhydrous MgSO_4_, and evaporated to dryness under reduced pressure. The crude product was purified through column chromatography on silica gel using hexane/chloroform (*v/v* = 5/1) as an eluent to afford compound **1**. (Yield: 80%).

#### 4.3.2. Synthesis of **2**

Compound **1** (8.9 mmol) and 4-bromoiodobenzene (2.97 g, 10.6 mmol) were dissolved in toluene (30 mL). After the solution was heated to 100 °C, CuI (0.15 g, 1.5 mmol) and KOH (1.23 g, 22.0 mmol) were added under N_2_ purge. The mixture was refluxed for 48 h at 120 °C. After being cooled, the mixture was washed with H_2_O (50 mL) three times, and the organic phase was dried over Na_2_SO_4_. After removal of the solvent, the residual was purified on a silica gel column with ethyl acetate/petroleum (*v*/*v* = 1/40) as the eluent, giving compound **2**. (Yield: 75%).

#### 4.3.3. Synthesis of **3**

A mixture of compound **2** (1.0 eq), (5-formylthiophen-2-yl) boronic acid (2.0 eq). Pd(dppf)Cl_2_ (10 mol%), and K_2_CO_3_ (5.0 eq) were added in mixed solvent (MeOH/toluene, *v*/*v* = 1/1). The reaction mixture was heated to 75 °C for 16 h. The reaction mixture was then filtered, and the solvent was removed. The residue was dissolved in DCM and washed using water. The combined organic layer was dried using MgSO_4_ anhydrous and then filtered. Solvent was removed and the residue was purified through silica gel chromatography, giving compound **3**. (Yield: 55%).

#### 4.3.4. Synthesis of **4**

The mixture of compound **4** (1.0 eq) and 2-(3,5,5-Trimethyl-2-cyclohexen-1-ylidene) propanedinitrile (2.0 eq) in dry EtOH was refluxed for 72 h at 78 °C. After being cooled, the solvent was removed, and the residual was purified on a silica gel column with dichloromethane/petroleum (*v*/*v* = 1/20) as the eluent. Pure AIEgens were obtained. (Yield: 87%).

### 4.4. Preparation of F-127 AIE Nanoparticles

Pluronic F-127 (4 mg) and TTTDM (1 mg) were dissolved in THF (1 mL), respectively, and then mixed uniformly after 2 min of sonication in a water bath. The mixtures of TTTDM and F-127 were dispersed in ultrapure water (10 mL). After the THF evaporated, the concentration of F-127 AIE nanoparcticles was obtained.

### 4.5. Detection of ROS Generation in Aqueous Solution

The commonly used DCFH–DA was used as an indicator to investigate the ROS generation efficiency of AIEgens in aqueous solution. The pre-activated DCFH–DA solution (DCFH, 40 μM) was added into the sample solution containing F-127 AIE NPs (0.2 μM). Afterward, the mixed solutions were irradiated using white light (20 mW cm^−2^) over different time intervals. The fluorescence of the indicator at 525 nm triggered by AIEgen-sensitized ROS was measured at the excitation of 488 nm.

### 4.6. Biofilm Culture

A single *P. aeruginosa* colony was transferred into 30 mL TSB medium followed by vigorous agitation. An amount of 100 µL inoculated medium was placed into each well of a 96-well plate then incubated at 37 °C for 7 days. A mushroom-like biofilm structure could be found on the bottom of the well. For fluorescence imaging and SEM experiments, the biofilms were cultured on cover glass using the same procedures.

### 4.7. Biofilm Dispersion Assay

An amount of 100 μL of the NPs at different concentrations was added into the wells. Simultaneously, we used PBS and NP solution as the control group. Then, the biofilms were incubated with the NPs for 6 h and, subsequently, irradiated with white light for 1 h. The cells were fixed with methanol and then stained with 200 μL of 0.3% crystal violet (CV) solution. After 30 min, the wells were vigorously rinsed at least four times with sterile PBS to remove unbound dye, and 200 μL of 33% acetic acid was added to release the dye. The biofilms were quantified using a microplate reader by measuring the absorbance after being treated or not treated with the liposomes at 590 nm. Each concentration of material was tested in five replicates, and three independent experiments were conducted for each group. Simultaneously, the viability of biofilm cells was counted using the plate count method.

### 4.8. Fluorescence Imaging

The fluorescence imaging of the bacterial biofilm treated with F-127 AIE NPs under white light irradiation was characterized through confocal laser scanning microscopy (CLSM). *P. aeruginosa* biofilms were incubated with PBS, and F-127 AIE nanoparticles were incubated with irradiation; then, a confocal microscope (Leica stellaris 8) was employed to observe the bacteria in the biofilm.

## Data Availability

Please contact the corresponding authors for the information.

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
