# Peer review of "Precise Molecular Engineering of Type I Photosensitizer with Aggregation-Induced Emission for Image-Guided Photodynamic Eradication of Biofilm"

_molecules, 2023, doi:10.3390/molecules28145368_

Round 1

Reviewer 1 Report

The authors report a novel type I photosensitizer TTTDM with the aggregation-induced emission property and its application to detection and eradication of bacterial biofilms. The authors demonstrated that its efficient reactive oxygen species generation and NIR fluorescence. Furthermore, TTTDM-encapsulating polymer nanoparticles can stain biofilms and ablate the labeled biofilm upon light irradiation. The reviewer thinks the contents are important. However, the provided data do not seem to fully support the conclusion. Therefore, the reviewer would strongly recommend that this manuscript would be reconsidered after addressing the following problems:

1.     The characterization data do not seem sufficient to identify each compound. (p. 3, l. 3 from the bottom) The authors stated that "The structure of the TTTDM and intermediated were fully characterized by 1H NMR, high resolution mass spectra (HRMS) in the Supporting Information (Figures S1–S2)". However, the reviewer could not find the corresponding spectra for the compounds 1 and 2, which may be firstly reported in this work as well. In addition, there are many unidentified peaks probably coming from impurities in the 1H NMR spectra of TTTDM and the compound 3 (2–0.5 ppm). The reviewer is worried about the effects of such impurities on the results. Furthermore, the reviewer highly recommends that 13C NMR spectra should also be provided.

2.     (p. 4, l. 6) The author mentioned that " HOMO electron densities of TTTDM was primarily distributed in methoxy-TPE and benzenamine moieties, while the LUMO was localized over the other thiophene and cyano fragments (Figure S3), indicating the separated HOMO-LUMO and typical D-A structural features." However, the reviewer could not find Figure S3 and any results and experimental details of calculations. Please provide them.

3.     (p. 4, l. 14) The author assigned the emission mechanism of TTTDM in solutions to TICT. Please provide further evidence supporting the TICT mechanism such as the result of excited-state calculations.

4.     Some typographical and grammatical errors: for example, (p. 5, l. 14) eradiation -> irradiation; (p. 6, l. 3) As show in -> As shown in; etc.

Author Response

Answers to Reviewer #1:

The authors report a novel type I photosensitizer TTTDM with the aggregation-induced emission property and its application to detection and eradication of bacterial biofilms. The authors demonstrated that its efficient reactive oxygen species generation and NIR fluorescence. Furthermore, TTTDM-encapsulating polymer nanoparticles can stain biofilms and ablate the labeled biofilm upon light irradiation. The reviewer thinks the contents are important. However, the provided data do not seem to fully support the conclusion. Therefore, the reviewer would strongly recommend that this manuscript would be reconsidered after addressing the following problems:

  1. The characterization data do not seem sufficient to identify each compound. (p. 3, l. 3 from the bottom) The authors stated that "The structure of the TTTDM and intermediated were fully characterized by 1H NMR, high resolution mass spectra (HRMS) in the Supporting Information (Figures S1–S2)". However, the reviewer could not find the corresponding spectra for the compounds 1 and 2, which may be firstly reported in this work as well. In addition, there are many unidentified peaks probably coming from impurities in the 1H NMR spectra of TTTDM and the compound 3 (2–0.5 ppm). The reviewer is worried about the effects of such impurities on the results. Furthermore, the reviewer highly recommends that 13C NMR spectra should also be provided.

Answer: Thank you so much for the constructive comments . We have supplemented the 1H NMR spectra of the intermediates in the Supporting Information. Meanwhile, we have further purified the TTTDM and confirmed its purity by 1H NMR and 13C NMR spectra.

  1. (p. 4, l. 6) The author mentioned that " HOMO electron densities of TTTDM was primarily distributed in methoxy-TPE and benzenamine moieties, while the LUMO was localized over the other thiophene and cyano fragments (Figure S3), indicating the separated HOMO-LUMO and typical D-A structural features." However, the reviewer could not find Figure S3 and any results and experimental details of calculations. Please provide them.

Answer: Sorry for the mistake. We have supplemented the data and details of the theoretical calculations in the revised Supporting Information.

  1. (p. 4, l. 14) The author assigned the emission mechanism of TTTDM in solutions to TICT. Please provide further evidence supporting the TICT mechanism such as the result of excited-state calculations.

Answer: Thank you for your suggestion. We have supplemented the results of excited-state calculations in the revised Supporting Information to support TICT effect of TTTDM.

  1. Some typographical and grammatical errors: for example, (p. 5, l. 14) eradiation -> irradiation; (p. 6, l. 3) As show in -> As shown in; etc.

Answer: Thank you for pointing out the mistakes we made. We have thoroughly revised the manuscript.

Reviewer 2 Report

This manuscript deals with a very important and current issue, which is a global health problem. However, the authors did not highlight the originality of the developed material in the manuscript. In fact, although it is a photosensitizer, others with similar activity are described in the scientific literature. Thus, it is suggested that the authors make clear the innovation presented in this manuscript.

Figures must be clear and self-explanatory. The Figures presented in this text are not comprehensible. It is suggested that the authors improve the figures by adding subtitles that make them clearer.

The authors do not present a conclusion of the work presented in the manuscript. The addition of work conclusion is important to journal readers.

The manuscript has English language errors, it is suggested that it be revised by a native speaker. 

Author Response

Answers to Reviewer #2:

  1. This manuscript deals with a very important and current issue, which is a global health problem. However, the authors did not highlight the originality of the developed material in the manuscript. In fact, although it is a photosensitizer, others with similar activity are described in the scientific literature. Thus, it is suggested that the authors make clear the innovation presented in this manuscript.

Answer: Thank you so much. In this work, we aimed to develop a design strategy for neutral type I photosensitizers (PSs) for biofilm eradication. Most of the reported type I photosensitizer probes applied in anti-biofilm were designed by ionization strategy. However, the charged PS molecules can interact well with both bacteria and biofilms without selectivity. The designed PS, TTTDM in this work was a neutral AIE probe and it can be encapsulated by a neutral polymer to form nanoparticles. The resulting nanoparticles selectively stained biofilm but not single bacterial cells. Furthermore, we have demonstrated that the TTTDM nanoparticles can be used to visualize biofilm growth by fluorescent imaging and eradicate biofilm through photodynamic therapy.

  1. Figures must be clear and self-explanatory. The Figures presented in this text are not comprehensible. It is suggested that the authors improve the figures by adding subtitles that make them clearer.

Answer: We have made modifications on the Figures.

  1. The authors do not present a conclusion of the work presented in the manuscript. The addition of work conclusion is important to journal readers.

Answer: Many thanks for your suggestions. We summarized the results in this work and presented a conclusion at the end of the manuscript.

Reviewer 3 Report

The manuscript entitled “Precise Molecular Engineering of Type I Photosensitizer with 2 Aggregation-Induced Emission for Image-Guided Photodynamic Eradication of Biofilm » is submitted by  R. T. K. Kwokac and B. Z. Tangabc  for publication in Molecules. The goal of this work was to design a Type-I photosensitizer with aggregation-induced emission property, used as TTTDM@ biocompatible polymer F-127 nanoparticles. These NPs (diameter around 100 nm) exhibit fluorescence in the NIR region and a quite efficient ROS generation through PDT, by enhancing intramolecular D-A interaction and confining molecular non-radiative transitions. Furthermore, the NPs can sufficiently penetrate biofilm matrix then image and eradicate bacterial biofilms upon white light irradiation.

Please complete references for “Photodynamic therapy (PDT) is an emerging clinical treatment method owing to its superior merits, such as spatial-temporal controllability, high efficiency, and noninvasiveness. PDT technique also shows promising treatment abilities towards drug-resistant bacteria and biofilms”, related to a review, J Photochem Photobiol B 2009, 96 (1), 1-8. More recent references could have been given J. Med. Microb. Diagn., 2014, 3, 4. Front. Microbiol. 2018, 9, 1299, Photochem. Photobiol. Sci. 2019, 18, 1020. Pharmaceutics, 2021, 13, 1995.

In conclusion, this manuscript is quite well written, of interest in the domain and deserve after asked complements) for publication in Molecules.

Author Response

Answers to Reviewer #3:

The manuscript entitled “Precise Molecular Engineering of Type I Photosensitizer with 2 Aggregation-Induced Emission for Image-Guided Photodynamic Eradication of Biofilm » is submitted by  R. T. K. Kwokac and B. Z. Tangabc for publication in Molecules. The goal of this work was to design a Type-I photosensitizer with aggregation-induced emission property, used as TTTDM@ biocompatible polymer F-127 nanoparticles. These NPs (diameter around 100 nm) exhibit fluorescence in the NIR region and a quite efficient ROS generation through PDT, by enhancing intramolecular D-A interaction and confining molecular non-radiative transitions. Furthermore, the NPs can sufficiently penetrate biofilm matrix then image and eradicate bacterial biofilms upon white light irradiation.

Please complete references for “Photodynamic therapy (PDT) is an emerging clinical treatment method owing to its superior merits, such as spatial-temporal controllability, high efficiency, and noninvasiveness. PDT technique also shows promising treatment abilities towards drug-resistant bacteria and biofilms”, related to a review, J Photochem Photobiol B 2009, 96 (1), 1-8. More recent references could have been given J. Med. Microb. Diagn., 2014, 3, 4. Front. Microbiol. 2018, 9, 1299, Photochem. Photobiol. Sci. 2019, 18, 1020. Pharmaceutics, 2021, 13, 1995.

In conclusion, this manuscript is quite well written, of interest in the domain and deserve after asked complements) for publication in Molecules.

Answer: Many thanks for your positive comments and suggestion. We have cited the recent reference on the research progress of photodynamic therapy in our revised manuscript.

Round 2

Reviewer 1 Report

All issues have been addressed. The reviewer thinks this manuscript is publishable in the present form.

Author Response

Our revised supporting information and manuscript are attatched with this email

  • Many reference citations in Introduction are missing ref. sources.
    Answer: Thank you so much for your comment. We have modified the reference source in the manuscript.
    - The yield of each compound is missing. Should be added to the Experimental section.
    Answer: Thank you so much for your comment. We added the yield of each compound in our experimental section.
    - The purity of each compound should be reported. For example, there are visible impurities in Compound 3 and TTTDM.
    Answer: Thank you so much for your comment. We reported the purity of each compound in our supporting information.
    - The NMR solvent is missing for all compounds.
    Answer: Thank you so much for your comment. We revised and added the NMR solvent in our supporting information
    - The chemical shifts are missing in the 13C NMR spectrum. The numbers are difficult to interpret in other NMR spectra. Please include the chemical shifts of each compound in the Experimental.
    Answer: Thank you so much for your comment. We revised and added the chemical shifts of each compound in supporting information
